# Corvids in Urban Environments: A Systematic Global Literature Review

**DOI:** 10.3390/ani11113226

**Published:** 2021-11-11

**Authors:** Isma Benmazouz, Jukka Jokimäki, Szabolcs Lengyel, Lajos Juhász, Marja-Liisa Kaisanlahti-Jokimäki, Gábor Kardos, Petra Paládi, László Kövér

**Affiliations:** 1Animal Husbandry Doctoral School, University of Debrecen, 4032 Debrecen, Hungary; paladi.petra@agr.unideb.hu; 2Arctic Centre, University of Lapland, 96300 Rovaniemi, Finland; jukka.jokimaki@ulapland.fi (J.J.); marja-liisa.kaisanlahti@ulapland.fi (M.-L.K.-J.); 3Department of Tisza Research, Institute of Aquatic Ecology, Centre for Ecological Research, Eötvös Loránd Research Network, 4026 Debrecen, Hungary; lengyel.szabolcs@ecolres.hu; 4Department of Nature Conservation Zoology and Game Management, University of Debrecen, 4032 Debrecen, Hungary; juhaszl@agr.unideb.hu (L.J.); koverl@agr.unideb.hu (L.K.); 5Institute of Metagenomics, University of Debrecen, 4032 Debrecen, Hungary; kg@med.unideb.hu

**Keywords:** adaptation, anthropogenic, Corvidae, habitat selection, life history trait, urbanization gradient

## Abstract

**Simple Summary:**

With regard to their high adaptability to human settlements and global distribution, corvid birds (crows, ravens, jays, etc.) are good models to understand the impacts of urbanization on wildlife. Here, we qualitatively reviewed the impacts of urbanization on corvids. At least 30 corvid species have become successfully accustomed or adapted to urbanized environments. The majority (72%, a total of 424 articles) of the studies reported positive effects of urbanization on corvids. The availability of easily accessible food and artificial nesting sites, coupled with low levels of predation, were found as the most important factors benefitting corvids in cities around the world. Studied topics ranged from population size and density, breeding biology and nesting site selection to control and management of Corvidae in cities. Despite biases in the distribution of the reviewed papers, our review attests that corvids have demonstrated high levels of adaptability to urban environments across space and time.

**Abstract:**

Urbanization is one of the most prevalent drivers of biodiversity loss, yet few taxonomic groups are remarkably successful at adapting to urban environments. We systematically surveyed the global literature on the effects of urbanization on species of family Corvidae (crows, choughs, jackdaws, jays, magpies, nutcrackers, ravens, rooks, treepies) to assess the occurrence of corvids in urban environments and the factors affecting their success. We found a total of 424 primary research articles, and the number of articles has increased exponentially since the 1970s. Most studies were carried out in cities of Europe and North America (45.5% and 31.4%, respectively) and were directed on a single species (75.2). We found that 30 corvid species (23% of 133 total) regularly occur in urban environments. The majority (72%) of the studies reported positive effects of urbanization on corvids, with 85% of studies detecting population increases and 64% of studies detecting higher breeding success with urbanization. Of the factors proposed to explain corvids’ success (availability of nesting sites and food sources, low predation and persecution), food availability coupled with diet shifts emerged as the most important factors promoting Corvidae to live in urban settings. The breeding of corvids in urban environments was further associated with earlier nesting, similar or larger clutches, lower hatching but higher fledging success, reduced home range size and limited territoriality, increased tolerance towards humans and increasing frequency of conflicts with humans. Despite geographic and taxonomic biases in our literature sample, our review indicates that corvids show both flexibility in resource use and behavioral plasticity that enable them to exploit novel resources for nesting and feeding. Corvids can thus be urban exploiters of the large-scale modifications of ecosystems caused by urbanization.

## 1. Introduction

Urbanization is a spatio-temporal process of the development of cities and the increase in the concentration of populations in them, followed by a transformation of natural habitats into artificial ones [1,2]. In general, urbanization is strongly associated with increased cover of imperious structures (e.g., buildings, streets) and human population density, as well as the fragmentation, degradation and loss of natural habitats. An urban development is an ecological modification that often alters the functions of a given ecosystem by affecting the structure of the food chain by removing or adding species, by encouraging human tolerance and adaptation, by increasing health risks for humans and wildlife and by modifying ecological processes in relation to ecosystem services [3]. Urbanization leads to complex, diverse systems characterized by high levels of human disturbance, pollution and landscape and environmental changes [1,2,4]. These changes can affect the biology, behavior, morphology and reproductive and survival traits of wildlife and can be responsible for the disappearance of native species and the appearance of non-native ones [5]. Therefore, understanding these effects is essential for successful wildlife conservation and management in urban habitats.

The negative impact of human-made landscapes and infrastructures on wildlife has been detected in many studies [6,7,8]. However, numerous studies have also described how certain species, such as corvids (e.g., crows, magpies), can benefit from these infrastructures, such as using buildings, poles and power lines as nesting sites [9,10,11]. In addition, anthropogenic food resources and milder microclimate in cities might benefit many corvid species [1,2,6]. Urbanization has been considered as an overwhelming evolutionary force acting on the life-history traits and population genetics of species [8]. Currently, urbanization is still expanding at an accelerating pace [12], unfortunately coinciding with a continuous increase in habitat loss. Although studies on the effects of urbanization on birds, at a community or individual species level, have been widely conducted, multi-species approaches with species belonging to the same family are still very scarce.

Corvidae is a family of mid to large-sized passerines. Many corvid species thrive in many types of urban environments, from the peripheral urban areas to highly urbanized urban core areas [13,14,15]. Because of the wide distribution areas of many corvid species and good adaptability to many habitats, corvid species are often described as urban adaptors and even exploiters [16,17,18]. Additionally, considering the high diversity and broad distribution of species of the Corvidae family, their spatio-temporal historical dispersal over numerous geographic and ecological areas may most likely contribute to the increase in taxonomic biodiversity [19]. Thus, corvids can be considered as the ideal subjects for investigating the effects of urbanization on birds. Lowry et al. (2013) have stated that the first observed adjustment shown by wildlife species in a human-made environment is a modification of behavior [20]. For example, numerous wild animals have been observed to alter their breeding, nesting and foraging patterns, diet composition, as well as vigilant behavior and vocalization, in response to human-made environments [21,22,23,24]. While some studies have explored the effects of urbanization on wildlife species and the adaptations of synanthropic birds (free-ranging wildlife living close to humans and benefiting from them) to urban habitats [5,16,25,26], individual studies tend to have a restricted geographical scale of inquiry, a single-species approach and usually focus only on a single trait that predicts adaptation [27,28,29,30].

In this paper, we systematically reviewed scientific papers to synthesize knowledge of corvid species in urban environments at a global scale. The main aims of the study were (i) to identify which corvid species have been studied in an urban context, and where and when they have been studied, (ii) to examine the effects of urbanization on corvid species by exploring how corvids respond to changes caused by humans, and how they evolve and survive in urbanized environments and (iii) to identify commonalities and differences between corvid species successfully colonizing urban environments around the world. We hypothesized that corvids are highly synanthropic and can adapt easily to modified environments. We predicted that corvids would display similar behavioral adjustments in different cities around the world, thanks to their high flexibility despite their different natural background. In addition, we aimed to better understand the reasons behind the successful colonization, establishment and adaptation of corvids to urban environments at a global scale. Finally, we aimed to explore the success or failure of management and conservation efforts associated with the corvids’ presence in cities worldwide. To our knowledge, this is the first comprehensive literature review specifically covering corvid species in urban areas around the world.

## 2. Materials and Methods

### Literature Search

We first searched for studies of corvid species by using the Scopus and the Web of Science databases and summarized and assessed the findings of studies relevant to urbanization in a global context. Our extensive and systematic literature survey followed the PRISMA guidelines and checklists for literature reviews (http://prisma-statement.org/, accessed on 26 November 2020, [31]). We used the following search script (search date: 30 September 2020): TS (topic) = (Corvids OR Crow OR Jay OR Magpie OR Nutcracker OR Brushcrow OR Chough OR Piapiac OR Raven OR Rook OR Treepie OR Jackdaw) and TS = (cities OR city OR urban OR suburban OR sub-urban OR town OR suburbs OR residential OR man-made OR human-made OR builtup OR buildup OR built-up OR build-up OR developed OR non-rural OR metropol) AND NOT (crowd OR crowe OR crown OR jaya) and restricting hits to the English language research papers and related chapters in books only.

In addition, we later searched the Corvids Literature Database [32], (search date: 17 October 2020) to check if our Web of Science and Scopus search missed any relevant publications. When using the Corvids Literature Database, we used the advanced search tools built in this database. We restricted our search to “City/Anthropogenic Impact” theme, “Normal paper” type of publications and papers from which at least an abstract was available. We did not make any search restrictions related to the year of publication or the language of publication. Therefore, our Corvids Literature Database search takes also into account non-English corvid publications that were not covered in our Scopus and Web of Science searches.

Overall, studies conducted over an urban-rural gradient (an order of settings based on the prevalence of human-mad infrastructures in association with human population density) or different urban habitats types, as well as comparative studies between urban and non-urban habitats, were included, since such studies can offer a wider perspective on urbanization’s impact on wild fauna and help to explore disturbance in urban environments [15,33,34,35].

The articles collected in the two searches were selected thoroughly by following the four phases of the systematic review flow diagram of the PRISMA guidelines. The PRISMA flow chart is given in Appendix A. To be included in the analyses, the article had to deal with any of the following topics in the context of corvid species.

1.Urban colonization, establishment, abundance, population dynamics and distribution, breeding ecology, nesting success, nest site selection, nestling development and growth.2.Territoriality and habitat use, activity, foraging, feeding, roosting behavior, diet availability and composition.3.Corvids’ tolerance and responses to human disturbance, human tolerance of corvids, and human-corvid conflict. We also included studies about specific biological or physiological parameters as these can be important to explain the birds’ responses to human-caused changes in their environment.4.Indirect influence of urban environmental or anthropogenic resources, such as the effect of anthropogenic food.5.Urban wildlife management and control.6.Pollution, metal contamination, disease transmission and zoonosis.7.Ecosystem services, such as seed dispersal.

Finally, we extracted the following information from the filtered studies: studied species, geographic location of study (continent, country), year of study and year of publication, for a better perspective on the evolution of the different bird populations in cities around the world.

The list of publications, with their basic information (*n* = 424), included in the analyses is given in Appendix A. The reference numbers in the tables refer to the individual publication numbers given in Appendix A.

## 3. Results

### 3.1. Corvid Publications

Our Scopus literature search found a total of 2623 articles, and Web of Science search found 2565 articles, of which 972 were duplicates (Appendix A). After screening and checking the eligibility of those articles, we ended up with 267 articles. Our Corvids Literature Database search found a total of 437 articles, from which 157 articles were usable after excluding duplicates (*n* = 57 dropped), screening (*n* = 338 dropped) and checking the eligibility (*n* = 135 dropped). Finally, our sample included 424 (267 + 157) articles for our analyses (heretofore referred to as “sample”). Additionally, 93.8% of the reviewed papers were English language publications; the additional non-English texts were mainly written in Japanese (2.36%), Polish (1.96%) and German (0.98%).

Urban corvids have been studied since the 19th century; however, most studies were published in the 21st century, with 46% of the articles published after 2010 (Figure 1).

Most studies were carried out in Europe or North America (45.5% and 31.4%, respectively; *n* = 424), followed by Asia and Australia (14.6% and 4.5%), and there were only a few studies from Africa (2.1%; Appendix A). Eight studies (1.9%) had a global study range. A total of 319 articles (75.2% of all 424 articles) were on a single species, 44 (10.4%) on two species, and 60 (14.2%) on more than two species, whereas 22 (5.2%) of the articles were community-level studies (Appendix A).

We found evidence of occurrence in urban environments in 30 species/subspecies/races of Corvidae (Appendix A). Most (76%, *n* = 424) of the reviewed studies considered the adaptation of crows and magpies to the urban environment. Additionally, population trends, breeding biology, nesting site selection, and human–corvid interactions have been often monitored. (Appendix A). The Eurasian Magpie (Pica pica; *n* = 93), the Rook (*Corvus frugilegus*), the Western Jackdaw (*Coloeus monedula*), the Hooded Crow (*Corvus corone cornix*), the American Crow (*Corvus brachyrhynchos*), the Scrub Jays (*Aphelocoma spps*.), the Common Raven (*Corvus corax*), and the Carrion Crow (*Corvus cororne corone*) were the most often studied Corvidae species. 

### 3.2. Effects of Urbanization on Corvids

From a total of 245 articles addressing the topic of the urban effect on corvid species, 177 articles (72.2% of the 245 articles) demonstrated positive effects of urbanization on the Corvidae, as various corvid species correlate positively with the proportion of built up establishments, demonstrating a continuous increase in population rates, and 42 articles (17.1%) reported negative effects, showing a decrease in the population density of the birds in question, and 26 articles (10.6%) reported no effect, reflecting thus no display of any changes in population trends (Figure 2). Specifically, the Eurasian Magpie, the Common Raven, the Rook, the Hooded Crow and the Carrion Crow have been reported to benefit from urbanization (Figure 2).

#### 3.2.1. Breeding Abundance/Density of Corvids

From 92 articles addressing the topic of the breeding abundance of corvid species in urban areas, 81 articles (88%) indicated that the abundance or density of corvid species increased with urbanization (Table 1). At least six corvids (the American Crow, the Common Raven, the Hooded Crow, the Carrion Crow, the Eurasian Magpie, the Rook and the House Crow (*Corvus splendens*) were unambiguously demonstrated to show an increase in abundance or density with increasing urbanization. In contrast, only a few species were observed to show a decrease with urbanization, and even in these species, the number of studies showing increases usually exceeded the number of studies showing decreases (e.g., the Western (also known as Eurasian) Jackdaw; Table 1).

#### 3.2.2. Breeding Success

We found 31 articles that studied the impact of urbanization on breeding success in ten corvid species (Table 2). The breeding success of the Eurasian Magpie, Common Raven and American Crow increased with urbanization, whereas opposite results were reported for the Scrub Jay (Table 2).

### 3.3. Factors Influencing Corvids

#### 3.3.1. Artificial Food

Fourteen corvid species were reported to use artificial food resources in urban habitats (Table 3). Ten species were reported to use garbage cans and dumps, and five species used bird feeders, while four species used carrion. The Magpie and the Carrion Crow were reported to use all of these kinds of artificial food resources (Table 3).

#### 3.3.2. Artificial Nest Site Use

Fourteen species were reported to use some kind of artificial nest site; of these species, 10 use exotic trees, 6 use poles and 5 use buildings, roofs and attics (Table 4). Specifically, the Common Raven uses poles, whereas the Eurasian Magpie uses exotic trees for nesting in urban areas (Table 4).

#### 3.3.3. Roosting within Urban Settings

A total of 23 articles reported that at least 11 corvid species use urban areas as nocturnal roosting areas (Table 5). Rooks, Eurasian Magpies and Western Jackdaws were most often reported to roost in urban settings (Table 5).

#### 3.3.4. Predation and Persecution

We found 30 articles that indicated either predation on or persecution of corvids (Table 6). Ten corvid species were reported as victims of predation events in 13 articles and, nine species were reported to be persecuted (Table 6). Many studies reported that the Eurasian Magpie suffers from predation and persecution.

#### 3.3.5. Human Related Infections and Contaminations

We found 17 articles reporting different kinds of viral, bacterial or fungal pathogens of human importance and 12 articles about the contamination of heavy metals in 9 corvid species (Table 7).

#### 3.3.6. Artificial Light and Noise

We found only six articles reporting on the responses of corvids to artificial light or noise in seven species. Artificial light was reported to negatively influence the House Crow and the Western Scrub Jay (*Aphelocoma californica*), but to positively influence the Rook, the Hooded Crow and the Western Jackdaw. Three species (Rook, Hooded Crow and Western Jackdaw) were reported to suffer from artificial noise.

### 3.4. Bird Responses to Human Developments

#### 3.4.1. Reproduction

Several articles demonstrated that breeding parameters, such as egg-laying date, clutch size, hatching and fledging success, were reported to differ between urban and rural corvid populations (Table 8). Fifteen studies of seven species demonstrated that egg-laying starts earlier in urban corvid populations than in non-urban populations. A total of 11 articles reported larger clutches in urban than non-urban corvid populations. A lower hatching success but a greater fledging success in urban than in non-urban corvid populations were reported in six and four articles, respectively. Nevertheless, four other articles have demonstrated lower fledging success in urban than in non-urban corvid populations. Such differences were most frequently reported in the Eurasian Magpie and the Florida Scrub Jay (*Aphelocoma coerulescens*), with eight and seven articles, respectively (Table 8).

Brood reduction, repeated nesting after nesting failure and nest reuse of the Eurasian Magpie, have been reported in four, two, and two studies, respectively. Brood reduction and repeated nesting were reported in one study each in the Florida Scrub Jay.

#### 3.4.2. Corvids’ Behavioral Responses to the Urban Environment

We found 26 articles reporting behaviors of 16 urban corvid species (Table 9). All 16 species were reported to show some kind of behavioral change or adjustment in response to the urban conditions, whereas only 3 species did not show such changes. The Common Raven was particularly often reported to change its behavior in urban settings.

#### 3.4.3. Corvids’ Responses to Human Presence

We found nine articles addressing the level of tolerance or vigilance in eight different corvid species. Seven out of eight species were reported to show greater tolerance towards humans, whereas only three species were reported to have an opposite reaction (Table 10).

#### 3.4.4. Human–Corvid Conflicts in Urban Settlements

We found 15 articles that described some kinds of corvid-human conflicts in urban areas (Table 11). Eight species were reported to cause noise, five species were reported to foul infrastructure and showed aggressive behavior, four species caused problems via fecal droppings and three species by scavenging on garbage (Table 11).

#### 3.4.5. Ecosystem Services of Urban Corvids

A total of 16 articles demonstrated the potential role of the corvids providing ecosystem services (Table 12). Seven species were identified as seed dispersers, and six species provided services with implications on human health. Corvids may be used as biosensors for different types of contaminations that can potentially result in the transmission of agents hazardous to humans, such as the West Nile Virus (WNV), a zoonotic viral infection often detected in crows and ravens [36,37].

## 4. Discussion

Our results indicate that corvids have been widely successful in adapting to urbanized environments over recent decades over a worldwide geographic scale. At least 23% of the 130 corvid species in the world are reported to live in urbanized environments, attesting this group’s extreme flexibility in resource use and highly opportunistic and plastic behavior, often enabling them to reach high abundances in various cities around the world. In contrast, we found no evidence of occurrence in urban environments for 100 corvid species. However, this does not mean that these species avoid cities and towns; rather, it reflects the lack of detailed studies documenting the occurrence and breeding of corvids in urban environments.

Accordingly, while the establishment and colonization of corvids in urban environments have drawn interest from researchers since the early 1990s, there has been a recent surge of scientific interest in corvids in urban environments in many areas of the world, and research on urban corvids is a rapidly emerging field of inquiry. Most of these studies address population trends, breeding biology and nest site selection in particular, in cities and towns, and a lower number of studies investigated the habitat use and movements, including home range estimates, of urban corvids. An overwhelming majority of the reviewed articles reported positive effects of urbanization on the population density or abundance of corvid species, and a majority reported higher breeding success. While several factors, such as nest site availability and low abundance of predators, have been suggested to explain the success of corvids in cities, easily accessible food sources, coupled with shifts in the diet of corvids seem to be the single most important factor promoting the success of corvids in cities. Many urban breeding corvid species start their breeding earlier, have larger or equal clutch sizes, lower hatching success, greater fledging success, reduced home range size, less territoriality and increased tolerance towards humans, as well as more conflicts with humans than their corresponding rural populations.

### 4.1. Corvids in Urban Environments: Change of Research Interests with Time

Most of the articles published in the early 1900s focused on the observation and description of corvid populations in the green spaces in cities of North America [38,39,40,41]. Similar studies were also conducted in Europe at this time [1,42,43,44]. One of the main goals of these early studies was to establish factors affecting the colonization of towns and cities by corvids, concentrating mainly on the availability of food sources and green spaces in the city. By the 1970s, with the continuous increase in corvids in urban areas, abundance censuses of the urban populations [45,46,47] and breeding ecology studies became the core subject of urban corvid research [48,49,50,51,52,53]. In the early1990s, the interest of researchers took a twisting turn to the rapid increase in human development around the world and its usually negative effect on wild fauna. Understanding how these synanthropic corvids adapt and benefit from urbanization became an important puzzle to be solved [54,55,56,57]. By the 21st century, earlier qualitative studies were supplemented by quantitative ones [58,59,60,61] that included experiments on the behavior of urban crows [62,63,64,65] and mathematical models to follow the movements of these birds in urban settlements [66,67,68,69].

### 4.2. Factors Explaining the Adaptation of Corvids to Urban Environments

Our literature search results indicated that most corvids are generalist species (able to consume a variety of foods and survive in different types of habitats) with a high degree of behavioral plasticity that makes them easily adaptable to environmental changes [14,70,71,72]. It has been widely established that many corvid species have flexible survival modes in order to benefit increasingly from anthropogenic habitats in different areas of the world. The most commonly assessed factor is food availability [73]. In every study assessing movement patterns and/or nest site selection, the impact of anthropogenic food sources on individuals’ choices was explored. For example, while it has been reported that non-breeding Common Ravens adapt their space use patterns to benefit from human food sources over large areas [74,75,76,77], breeding pairs choose to nest in areas with good food availability nearby their nests [61,77,78]. Indeed, earlier studies have indicated that breeding Ravens forage mostly around their nest [79,80]. Moreover, movement patterns of the American Crow are strongly correlated with the abundance of anthropogenic resources [71,81,82], as easily accessible food in urban areas was described as the main cause for the regular movement of rural American Crow fledglings to the cities, thus resulting in annual increases of the urban populations [71,82,83]. Furthermore, food availability highly influenced the abundance and habitat choice of urban Carrion Crows and Jungle Crows (*Corvus macrorhynchos*) [84,85,86,87,88]. Even the Alpine Chough (*Pyrrhocorax graculus*), one of the least explored corvids, was observed to alter its foraging behavior and to increase its survival based on the availability of anthropogenic foods [55,70,89]. Winter bird feeding in cities and towns by humans also further increases foraging opportunities to corvids, such as for the Eurasian Magpie and Hooded Crow in northern latitudes [90]. The role of food availability is also supported by indirect evidence; for example, the decline of the Western Jackdaw populations in several European cities is partially caused by the lack of food [91,92].

In contrast, some studies suggested that the role of anthropogenic food availability in explaining the corvid colonization of cities could be overrated. Vuorisalo et al. (2003) suggested that while easily available food is certainly important, the local overabundance of food does not explain the sudden increase in the Hooded Crow population density in the 1960s in Finland [93]. Instead, they suggested that less persecution by hunting and the absence of natural predators, as well as the availability of novel nesting opportunities in the city also played an important role in this expansion. In addition, studies on the Little Raven (*Corvus mellori*), in Australia reported that variation in home range size was mainly related to the habitat type associated with the distribution of food sources [72,94,95].

The second most frequently mentioned factor influencing corvid presence in urban habitats is the availability of novel nesting sites in the cities, which, along with their highly behavioral flexibility, can also explain the colonization of cities by corvids. For example, the Common Raven is almost exclusively nesting on anthropogenic structures, such as electric poles, in cities [3,74,96,97]. Magpies and Hooded Crows were also reported to change their nesting habits in urban settings. For example, Magpies tend to nest on the highest trees available in response to high levels of disturbance [5,27,98,99]. However, with decreased persecution, Magpies can nest on a less preferred site and build their nests at lower heights [17]. Hooded Crows will also nest on non-preferred tree species and at lower heights as the urban population increases and preferred nesting sites become scarce [100]. Again, cavity nesters, such as the Western Jackdaws offer indirect evidence for the importance of nesting sites because jackdaw populations declined in some European cities mainly due to the loss of suitable nest sites caused by renovations and modernizations of buildings that had been previously suitable for cavity-nesting [91,100,101,102].

Additionally, access to better nesting and feeding resources in the cities often translates into shifts in reproductive behaviors. For example, Scrub Jays in urban areas start breeding about three weeks earlier than Scrub Jays in rural areas, mostly due to the higher availability of food [27,103]. Bagyura et al. (2017) also observed unusually early breeding of Common Ravens nesting on electric poles in Hungary [30]. Nesting earlier does not necessarily translate into higher success; for example, earlier breeding in the Alpine Chough resulted in an increase in nest failure in urban settlements [7].

The third most frequently mentioned factor influencing urban corvids is their high adaptability in behavior, physiology and breeding biology. Corvids’ responses to environmental change have been shown to be highly flexible, and it is suspected to be correlated with specific biological traits shared by corvid species [21,22,28,53,104,105,106]. For instance, breeding biology parameters of several corvid species were reported to differ between urban and non-urban populations in several corvid species, including the Eurasian Magpie, American Crow, Common Raven and Hooded Crow, and to a lesser extent, the Steller’s Jays (*Cyanocitta stelleri*) and Jackdaws. In addition to earlier breeding, urban birds produce smaller clutches (fewer eggs) and rear their young longer until fledging, as well as produce smaller fledglings and fewer potential future breeders than non-urban birds [3,21,24,30,107]. While some of these differences appear to imply disadvantages of nesting in urban environments, they did not seem to affect the size or growth of the urban populations of these species. This discrepancy may be explained by two factors. First, the increasing population sizes of some urban corvids may be due to the immigration of young crows from adjacent rural populations into the city [81], as the higher breeding success in rural areas induced movements of rural dispersers into the city, creating a net immigration of crows into cities [82]. Second, Marzluff et al. (2001) demonstrated that urban crow populations tend to increase partially because survival rates are higher in urban than in rural corvid populations in North America [81]. It is important to note that most studies carried out during non-breeding seasons suggested that chances of survival are probably higher in cities than in non-urban habitats, as attested by large numbers of individuals that move from rural areas to cities for overwintering, at least in the Northern Hemisphere [108,109,110,111].

In addition, some corvids can adapt to changing light and noise conditions caused by urbanization. For example, Ciach and Frohlich (2017) indicated that the density of wintering Rooks and Magpies in southern Poland increased with increasing artificial light, but decreased with increasing noise [112]. Nevertheless, Ciach and Frohlich (2017) also pointed out that food availability during winter is probably the primary factor explaining corvid densities in urban areas [112]. It is plausible that artificial lights in cities may increase the time that the corvids can spend foraging, as has been observed in the case of urban pigeons [113].

### 4.3. Corvids’ Responses to Urbanization

Since access to anthropogenic resources in the cities and high levels of adaptation to novel environments often translate into shifts in corvids’ activities, as stated above, many corvids change their behavior and get accustomed to human presence. The birds’ responses to human presence and proximity have been extensively studied, often in experiments, in the American Crow, the Carrion Crow, Torresian Crow (*Corvus orru*) and Jackdaws. The various behavioral experiments aimed to explore the crows’ tolerance towards humans and their danger recognition ability in urban settlements [29,89,114], while other studies aimed to investigate social learning in these birds [115,116,117,118]. Decreased persecution in cities in the last few decades as opposed to rural areas appears to be an important factor promoting the corvids’ tolerance and habituation to humans and traffic; this tolerance is a pre-requisite of colonization of city centers as breeding habitat [17,93]. Moreover, such habituation has been widely documented in the American Crow [71,81,119].

In addition, several studies demonstrated that corvids can depend on social cues to learn about dangers in a given area and that they are able to communicate this information to each other [65,75,120]. An extreme case of habituation to human presence has been observed in a Carrion Crow captured in Vienna Zoo, which did not show any sign of struggle during handling by researchers [121]. Finally, studies based on measurements of crows’ flight initiation distance upon human approach have indicated that escape distances are considerably shorter in urban than in non-urban areas in many corvid species [122,123].

### 4.4. Corvid Human Interactions

The increased abundance of corvids due to urbanization means that corvid-human interactions will also increase. Corvids have been reported to cause conflicts due to their disturbing noise, fecal droppings, garbage scattering, damages to infrastructures and aggressive behavior towards humans and pets. For that reason, corvids have often been perceived as nuisance birds and have intensively been persecuted or even hunted in many cases [105,124,125]. However, in relation to wildlife and nature conservation, for example, the EU Birds Directive [126], the disturbance of birds during their spring migratory and breeding periods is banned, as well as prohibiting the large scale and non-selective means of bird killing.

Because corvids often forage on communal waste found in household garbage dens, on commercial refuse in dumpsters, near outdoor restaurants and food stands, trashcans in parking lots, and landfills [82,119,127], corvids can serve as vectors of disease transmission. In particular, communal roosts and the feeding location of corvids in cities greatly increase the chances of disease transmission, which is of great concern for human and animal health [128,129]. Several studies addressed the potential role of wild birds as vectors and spreaders of pathogens of important zoonotic and other human-related diseases in urban areas. Most studies of zoonosis in crow species focused on the west Nile virus (WNV), which can cause high mortality in corvids, as demonstrated in the American Crow [36,37,130,131,132]. Due to their high susceptibility to WNV, crows and ravens can thus be used as biosensors or early indicators of the presence of this virus in a given area. Similarly, corvids have been proposed or already been used to detect other pathogens of public health concern [133,134,135,136].

In addition to diseases, contamination by human-made pollutants has been documented worldwide in various urban corvids. For example, increased lead concentrations and high levels of dioxins have been detected, and other environmental chemicals have been observed in Eurasian Magpies [137,138,139,140,141], Common Ravens [142], Rooks [143,144,145] and Jungle Crows [146,147,148]. While these studies emphasize the detrimental impacts of these pollutants on urban wildlife, their biological and physiological implications on the survival or reproduction of wild animals in urban areas are not yet fully understood. Finally, high corvid abundance in cities often leads to the homogenization and/or depauperating of the urban bird fauna [149,150]. Some studies have indicated that the expansion of corvids, or their increase, in urban habitats have caused decreases in the richness or abundance of other species mainly via increased nest predation rate [25,151,152].

Based on the artificial nest predation experiments, corvids are often perceived as efficient nest predators that directly impact the populations of other bird species [153,154]. Despite this perception, relatively few experimental removal or population control studies have been performed in corvids [151,155,156,157]. Most of these studies found that the overall impact of corvid species on nest predation rate of other species is rather small, and that the populations of other bird species are less likely to be limited by corvid predation than suspected before. Predation by corvids is thus likely an effect that influences other bird species in synergy with other negative impacts on those species. Finally, although corvids are often perceived as a nuisance, recent studies shed light on their potential role to provide supporting ecosystem services, such as helping seed dispersal of fugacious tree species through transporting fruits and seeds per caching trips over long distances [158,159], and regulating ecosystem services, such as health-based services by acting as a surveillance tool of important zoonotic pathogens and by reducing animal remains by scavenging within urban settings [23,36,37,160,161].

### 4.5. Management Efforts

Corvids in urban areas have often been considered as pests and sources of nuisance, thus have become the target of management efforts around the world [125,162,163]. Human-corvid conflicts emerge because of the corvids’ garbage scattering, fouling infrastructures, roosting in high numbers on roofs and in parks, unpleasant vocalization, attacks on pets and humans during chick-rearing, use of sensitive infrastructure for nesting and predation on birds and other animal species dear to humans. These conflicts have initiated a large number of studies discussing the management of these birds in urban areas [164,165,166,167,168]. Many studies focused on the House Crow, a highly invasive corvid originally from Southeast Asia (mainly Pakistan and India) that has recently colonized and been thriving in cities of the sub-Saharan region and in the Middle East, and on the Pied Crow (*Corvus albus*) in Brazil. The wide range of these birds and their high flexibility makes them targets to persecution by shooting [26,125,162,163,164]. Other examples of management include the destruction of Chihuahuan (*Corvus cryptoleucus*) and Common Ravens nests on electric poles [156,169,170], scaring away winter roosts of American Crows in U.S. cities [124,165] and trapping and removal of Hooded Crow and Carrion Crow individuals in cities in Europe [166,167,170]. However, the current population status of these species indicates that the success of these management efforts so far is limited. This implies that controlling the resources vital for urban corvid populations may be more successful at reducing their populations than direct population control. For example, Chong et al. (2012) believe that efficient waste management contributed largely to the reduction in the House Crow population in Singapore [26]. Restani, Marzluff and Yates (2001) also demonstrated the role of controlling access to food sources on corvid population growth in North America by focusing on the necessity of increased attention to garbage storage, animal husbandry practices and bird feeding around residential areas [75].

### 4.6. Study Limitations

Our review has geographical and taxonomical biases that must be considered in the correct interpretation of the patterns reported. The studies in our literature sample had some geographical biases, with the majority of studies coming from the Northern Hemisphere, North America and Europe in particular. This was mainly because of a language bias, as even though we were able to assess materials published in different languages, most corvids’ studies were published in English. Yet, it is important to note that only foreign papers with an English abstract were accepted for this review. Nevertheless, we found no substantial differences in the effect of urbanization on the reviewed corvid species from the different cities around the world. Although these cities and towns represent various biogeographical and environmental characteristics, anthropogenic changes in urban environments are rather similar worldwide, which increases the external validity of our results.

The geographical bias also leads to a taxonomic (species) bias in our sample because corvid species in our literature sample are limited to those species that are distributed in the areas where studies were conducted and published in English. With numerous studies carried out in Europe and North America since the beginning of the previous century, we found only nine and two studies on the colonization/invasion of a non-native corvid (House Crow) in sub-Saharan Africa and Latin America, respectively. In addition, some species have been the object of numerous management studies, especially the House Crow [10,125,162,164,171], the American Crow [124,165,172], the Eurasian Magpie [138,173,174] and, to a smaller extent, the Common Raven [76,156], the Hooded Crow [100,168,175] and the Rook [168]. Studies of management have reported on the use of different methods ranging from direct shooting [171], trapping [14,100,175], egg removal [164], poisoning [164], pruning roost trees [125], acoustic scaring of birds [124,176], dispersing roosts with lasers [165] and refuse management in cities [177].

In addition, by following the PRISMA guidelines, we may have missed some publications in relation to our topic. For example, bird community studies broadly including corvid species might not have been fully detected. This could potentially be an additional shortcoming of our analysis. While many of our results reported are from the given subset of corvid species, the patterns found and the conclusions drawn will likely be interpretable and useful for researchers studying species other than the well-studied ones.

### 4.7. Research Needs and Avenues

Our review also identified several gaps or shortcomings in our knowledge and thus suggests new avenues for research. First, we found a substantial geographical and taxonomical bias in studies of urban corvids (see above), and there is a dire need to initiate or report results from studies conducted in little-known geographic areas and corvid species, e.g., the Little Raven, the Chihuahuan Raven and the Jungle Crow. In the case of rapidly spreading species, such as the House Crow, there is a need to know more about the biology of species in their native range as opposed to their invaded, non-native range to understand the factors influencing their success and to provide information for their management.

Second, while we found a diversity of methods applied to study corvid species, certain species were studied by a limited set of methods. Our sample includes studies applying many different methods, such as: (i) experimental studies of corvid behavior to human presence or of responses of corvid populations to management efforts, such as removal experiments for predation impact studies, (ii) spatial methods, such as transect surveys and radio/satellite tracking methods, for home range and space use studies [14,178,179] and (iii) correlative studies to collect biological and ecological parameters using existing data over time and space (e.g., [180,181]), which can provide informative comparisons of urban and rural populations of corvid species to understand the factors facilitating the colonization and increase in corvids in urban environments. This methodological diversity was not independent of species; for example, the Eurasian Magpie was the most studied Corvidae, mostly because it is native to Europe, and was most often studied by correlative methods primarily to understand nest site selection. Similarly, studies of American Crows and Common Ravens were mostly initiated for understanding the role of anthropogenic food sources and for monitoring of West Nile Virus. Moreover, while only a few studies explored the biological aspects of urban House Crows in their native range, numerous studies discussed management tools and methods to eradicate this bird in their non-native areas. While the methodological approaches will lead to interesting results on their own, we stress that their combined use can probably further increase the applicability of results for both science and practice.

Third, we need to know more about whether and how exposure to pollutants widespread in the urban environments affects the biology or physiology of corvids. Pollution is likely to have a harmful impact on all wildlife, but this likelihood probably varies between species and by the type or concentration of the pollutant.

Finally, we need more information on management and conservation. Management of crow populations that have exceeded the acceptable level of abundance has become a necessity in many parts of the world. However, we have little information on successful long-term methods of population control and removal of corvids from urban environments. While direct methods, such as hunting bag data, in the case of control by shooting can be informative in many cases, indirect methods, such as control of resources (e.g., reducing food availability), are much less frequently studied or reported, and thus little is known on the efficiency of these approaches. More studies are needed to understand the presumed negative impact of corvids on assemblages of urban species of birds and other taxa (e.g., rodents) for future management of corvids, for the conservation of urban wildlife and for city planning. One interesting study topic might be why some corvid species are urbanized, whereas others, such as the Siberian Jay (*Perisoreus infaustus*), are not. Evidently, the habitat needs of some corvid species are not fulfilled in cities. For example, the Siberian Jay is an old-growth bird species living in old-aged and large-sized coniferous forests [182]; therefore, this species does not breed or over-winter in cities. Since urban green areas are normally quite small-sized, fragmented and deciduous tree-dominated [183].

## 5. Conclusions

Our review shows that corvids have long been associated with the development of urban environments, and their worldwide distribution makes them the perfect model system to study the effects of urbanization on wildlife. A considerable proportion of species in the Corvidae family have already been shown to adapt to urban environments, and, with consideration to the geographical and taxonomical bias in our literature sample, it appears likely that many more species will be shown to be successful in the adaptation process. The primary traits of corvids that enable them to exploit new, urban environments are their high behavioral plasticity and flexible resource use. With easily accessible food being the most important resource attracting these birds into cities; influencing different traits of habitat selection (e.g., use of new nesting sites) and life history (e.g., earlier nesting, larger clutches, higher fledging success, reduced home ranges and territoriality), as well as behavior (increased tolerance of humans); understanding the relative importance of these changes in each species will be fundamental to better understand the adaptation process and human–wildlife interactions and to develop efficient management applications. While the effects of urbanization on numerous corvid species have been relatively well explored, there are important gaps in our knowledge, calling for a more diversified approach to study this process with different, complementary methods and a focus on the potential benefits of this process, such as the ecosystem services that corvid species provide. We encourage researchers to also address these aspects for a more balanced view of corvids in urban environments.

## Figures and Tables

**Figure 1 animals-11-03226-f001:**
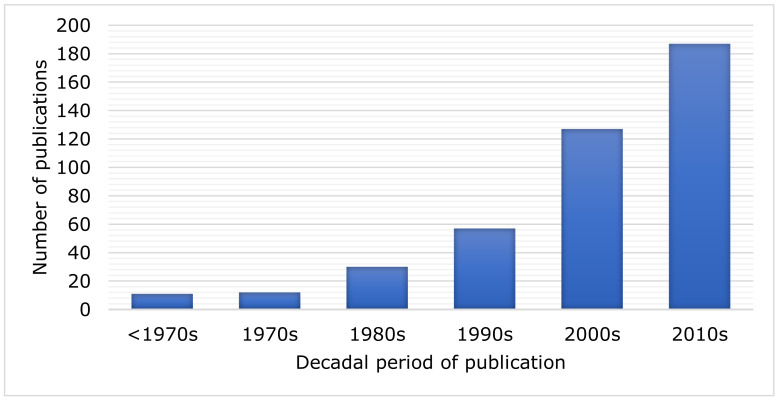
Number of articles published on Corvidae species in urban environments by decade (*n* = 424).

**Figure 2 animals-11-03226-f002:**
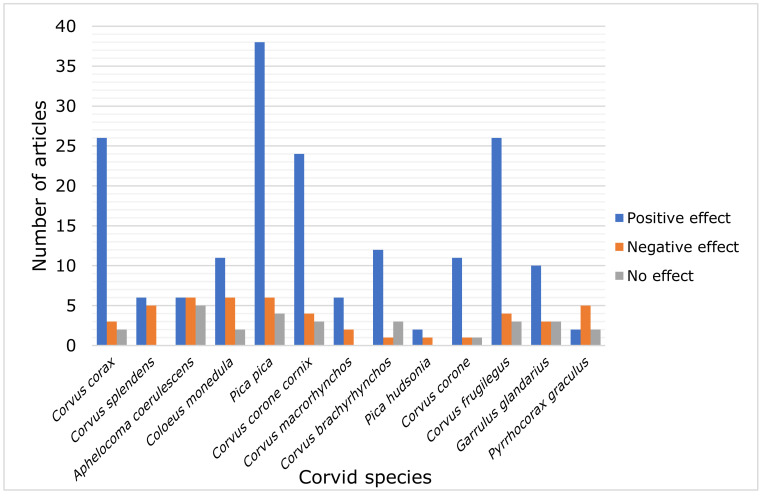
Number of articles reporting positive, negative or no effect of urbanization on the most frequently studied corvid species (*n* = 245).

**Table 1 animals-11-03226-t001:** Articles reporting positive, negative or no effects of urbanization on the abundance/density of populations in different corvid species (total *n* = 92). The reference numbers in the table refers to the individual publication numbers given in Appendix A.

Species	*N*	Population Decrease	Population Increase	No Change
Alpine Chough (*Pyrrhocorax graculus*)	2	334		303
American Magpie (*Pica hudsonia*)	1		262	
American Crow (*Corvus brachyrhynchos*)	11		91, 193, 244, 245, 255, 268, 271, 272–273, 305, 380, 381	
Australian Raven (*Corvus coronoides*)	1		296	
Blue Jay (*Cyanocitta cristata*)	4		91, 221, 305, 331	
Carrion Crow (*Corvus corone*)	9		179, 189, 295, 357, 364, 371, 388, 406, 409	
Common Raven (*Corvus corax*)	10		189, 193, 200, 201, 202, 242, 245, 249, 271, 297, 366	
Eurasian Magpie (*Pica pica*)	13	401	64, 130, 137, 138, 157, 171, 172, 176, 179, 186, 189, 192, 234, 237, 242, 249, 250, 251, 252, 335, 423, 340, 382, 397, 398	252
Eurasian Jay (*Garrulus glandarius*)	4	242, 249	250, 335	
Fish Crow (*Corvus ossifragus*)	2		91, 258	
Hooded Crow (*Corvus cornix*)	11		157, 179, 185, 189, 207, 209, 242, 249, 250, 251, 252, 330, 335, 350, 357, 364. 365, 398, 418, 423	252
House Crow (*Corvus splendens*)	5	269	157, 306, 356, 377	
Jungle Crow (*Corvus macrorhynchos*)	5		190, 356, 388, 406, 407	
Red-billed Choughs (*Pyrrhocorax pyrrhocorax*)	3	36		175, 285
Oriental Magpie (*Pica pica sericea*)	2		111, 188	
Pied Crow (*Corvus albus*)	2		206, 231	
Rook (*Corvus frugilegus*)	15	397, 398, 251, 281	242, 249, 141, 186, 189, 249, 250, 275, 279, 294, 308, 335, 395, 410, 418, 419, 420	417
Steller’s Jay (*Cyanocitta stelleri*)	4	262, 361		245, 271
Western Jackdaw (*Coloeus monedula*)	14	22, 47, 64, 86, 310	242, 249, 250, 251, 335, 397, 398	189, 252

**Table 2 animals-11-03226-t002:** Articles reporting lower, higher or no effects on breeding success in urban areas relative to rural areas in different corvid species (*n* = 31). The reference numbers in the table refer to the individual publication numbers given in Appendix A.

Species	*N*	Breeding Success In Urban Areas		
		Lower	Higher	Unchanged
American Crow (*Corvus brachyrhynchos*)	5		245, 246, 255, 271, 272, 273	
Common Raven (*Corvus corax*)	4		19, 211, 212, 245, 372	
Carrion Crow (*Corvus corone*)	2	300		299
Eurasian Magpie (*Pica pica*)	8	110	14, 137, 138, 170, 171, 341, 385	
Hooded Crow (*Corvus cornix*)	1	396		
House Crow *(Corvus splendens*)	1	6		
Rook (*Corvus frugilegus*)	1		120	
Scrub Jay (*Aphelocoma spp.*)	6	274, 316, 324, 360	4, 5	
Steller’s Jay (*Cyanocitta stelleri*)	2		246	245
Western Jackdaw (*Coloeus monedula*)	2	11, 260		

**Table 3 animals-11-03226-t003:** List of articles in relation to artifical food resource use of corvids in urban environments (*n* = 28). The reference numbers in the table refer to the individual publication numbers given in Appendix A.

Species	*N*	Waste Food (Garbage Cans, Dumps)	Arthropods or Seeds	Bird Feeders	Carrion
Alpine Chough (*Pyrrhocorax graculus*)	2	65, 303		65	
American Crow (*Corvus brachyrhynchos*)	2	63, 369			
Australian Raven (*Corvus coronoides*)	2				277, 313
Carrion Crow (*Corvus corone*)	4	18, 65		65	164, 322
Common Raven (*Corvus corax*)	3	26, 37, 75		26	
European Jay (*Garrulus glandarius*)	1			148	
Eurasian Magpie (*Pica pica*)	8	18, 65, 71, 78	78	65, 113, 148	164, 322
Jungle Crow (*Corvus macrorhynchos*)	3	404, 407, 411			
Little Raven (*Corvus mellori*)	1				30
Pied Crow (*Corvus albus*)	1	62			62
Rook (*Corvus frugilegus*)	4	18, 71, 279	419		
Scrub Jay (*Aphelocoma sp.*)	3	65, 78	78	65, 312	
Torresian Crow (*Corvus orru*)	1		72		
Western Jackdaw (*Coloeus monedula*)	1	18			
White-naped Jay (*Cyanocorax cyanopogon*)	1	21	21		

**Table 4 animals-11-03226-t004:** List of articles in relation to artificial nest site selection (*n* = 47). The reference numbers in the table refer to the individual publication numbers given in Appendix A.

Species	*N*	Artificial Poles	Exotic Trees	Buildings: Roofs, Attics
Alpine Chough (*Pyrrhocorax graculus*)	2		65	36
American Crow (*Corvus brachyrhynchos*)	1			288
Australian Raven (*Corvus coronoides*)	1		224	
Black-billed Magpie (*Pica hudsonia*)	3	111	112, 188	
Carrion Crow (*Corvus corone*)	4	163	65, 415	415
Common Raven (*Corvus corax*)	8	2, 27, 76, 77, 163, 333, 376		61
Eurasian Magpie (*Pica pica*)	15	401	12, 65, 136, 169, 180, 270, 278, 367, 368, 408	38, 66, 367, 368
Hooded Crow (*Corvus cornix*)	4		205, 256, 278, 408	
House Crow (*Corvus splendens*)	5	106	6, 107, 327	8
Jungle Crow (*Corvus macrorhynchos*)	3		415	405, 415
Little Raven (*Corvus mellori*)	1	39		
Pied Crow (*Corvus albus*)	2	84, 241		
Rook (*Corvus frugilegus*)	3		142, 278	66
Scrub Jay (*Aphelocoma sp*.)	4		65, 95, 96, 105	
Western Jackdaw (*Coloeus monedula*)	4		309	11, 47,66, 86

**Table 5 animals-11-03226-t005:** List of articles reporting roosting corvids in urban areas (*n* = 23). The reference numbers in the table refers to the individual publication numbers given in Appendix A.

Species	*N*	Observations of Roosts
American Crow (*Corvus brachyrhynchos*)	3	132, 328, 369
Common Raven (*Corvus corax*)	3	102, 115, 392
Eurasian Magpie (*Pica pica*)	5	102, 135, 161, 336, 382
Grey Crow (*Corvus tristis*)	2	307, 357
Hooded Crow (*Corvus cornix*)	1	336
House Crow (*Corvus splendens*)	2	16, 289
Northwestern Crow (*Corvus brachyrhynchos caurinus*)	1	195
Pied Crow (*Corvus albus*)	1	206
Rook (*Corvus frugilegus*)	8	60, 102, 165, 166, 254, 336, 378
Torresian Crow (*Corvus orru*)	1	119
Western Jackdaw (*Coloeus monedula*)	5	60, 102, 135, 146, 336

**Table 6 animals-11-03226-t006:** List of articles reporting predation events and persecution towards to corvids in urban areas (*n* = 30). The reference numbers in the table refer to the individual publication numbers given in Appendix A.

Species	*N*	Predation	Persecution
American Crow (*Corvus brachyrhynchos*)	7	65, 127, 369	73, 127, 133, 134
Carrion Crow (*Corvus corone*)	5	65	357, 364, 388, 409
Choughs (*Pyrrhocorax spp*.)	3	65, 334	334
Common Raven (*Corvus corax*)	2		115, 326
Hooded Crow (*Corvus cornix*)	10	205, 256, 396	73, 210, 229, 325, 357, 364, 365
House Crow (*Corvus splendens*)	7	106, 107	54, 70, 106, 107, 121
Rook (*Corvus frugilegus*)	4	213, 350	210, 325
Steller’s Jay (*Cyanocitta stelleri*)	2	65, 361	
Western Jackdaw (*Coloeus monedula*)	6	65, 309, 350	226, 229, 325

**Table 7 animals-11-03226-t007:** List of articles about human-related infection and contaminations of corvids (*n* = 29). The reference numbers in the table refers to the individual publication numbers given in Appendix A.

Species	*N*	Heavy Metal Contamination	Zoonotic Pathogens
American Crow (*Corvus brachyrhynchos*)	6		183, 220, 235, 236, 374, 386
Common Raven (*Corvus corax*)	1		232
Eurasian Magpie (*Pica pica*)	5	99, 100, 393, 394	225
Hooded Crow (*Corvus cornix*)	1		387
House Crow (*Corvus splendens*)	3	158, 167	259
Jungle Crow (*Corvus macrorhynchos*)	3	204,	286, 389
Little Raven (*Corvus mellori*)	1		34
Rook (*Corvus frugilegus*)	9	238, 282, 283, 284, 417	43, 390, 311, 363
Western Jackdaw (*Coloeus monedula*)	1		311

**Table 8 animals-11-03226-t008:** Number of articles reporting differences between urban and rural areas in egg-laying date, clutch size, hatching success and fledging success in ten corvid species (*n* = 31). The reference numbers in the table refer to the individual publication numbers given in Appendix A.

Species	*N*	Laying Time		Clutch Size			Hatching Success		Fledgling Success	
		Earlier Date	Normal Time	Normal Size	Larger Size	Smaller Size	Lower Success	No Difference	Lower Success	Higher Success
American crow (*Corvus brachyrhynchos*)	1			255				255		255
California scrubjay (*Aphelocoma californica*)	1			360			360			
Carrion crow (*Corvus corone*)	2								299, 300	
Common raven (*Corvus corax*)	4	19, 212		19						211, 212, 372
Eurasian Magpie (*Pica pica*)	10	14, 110, 138, 170, 171, 341, 344	385	14, 138, 170, 341			159, 341			14, 136, 137, 170, 171, 341, 385
Florida scrubjay (*Aphelocoma coerulescens*)	6	123, 316, 320, 324			5, 316, 320		4, 5, 324			
Hooded crow (*Corvus cornix*)	2			300, 396						
House crow (*Corvus splendens*)	1	6				6	6		6	
Rook (*Corvus frugeligus*)	1	120								120
Western Jackdaw (*Coloeus monedula*)	3		11	11, 359		260	359		260	

**Table 9 animals-11-03226-t009:** List of articles reporting behavioral change or adjustment of corvid species in urban environments (*n* = 26). The reference numbers in the table refer to the individual publication numbers given in Appendix A.

Species	*N*	Behavioral Change or Adjustments	No Behavioral Change or Adjustment
American Crow (*Corvus brachyrhynchos*)	3	271, 380	92
Carrion Crow (*Corvus corone*)	2	94, 140	
Common Raven (*Corvus corax*)	11	75,76,77, 115, 147, 149, 212, 233, 271, 297	92
Eurasian Jay (*Garrulus glandarius*)	1	140	
Eurasian Magpie (*Pica pica*)	3	13, 140, 159	
Hooded Crow (*Corvus cornix*)	1	160	
House Crow (*Corvus splendens*)	1	228	
Jungle Crow (*Corvus macrorhynchos*)	2	151, 152	
Little Raven (*Corvus mellori*)	3	362, 375	375
Pied Crow (*Corvus albus*)	1	247	
Choughs (*Pyrrhocorax spp*.)	1	173	
Rook (*Corvus frugeligus*)	2	140, 166	
Steller’s Jay (*Cyanocitta stelleri*)	1	271	
Torresian Crow (*Corvus orru*)	1	119	
Western Jackdaw (*Coloeus monedula*)	2	140, 160	
White-necked Raven (*Corvus albicollis*)	1	247	

**Table 10 animals-11-03226-t010:** List of papers about corvids’ tolerance towards humans (*n* = 9). The reference numbers in the table refer to the individual publication numbers given in Appendix A.

Species	*N*	High Tolerance or Low Vigilance	Low Tolerance or High Vigilance
Alpine Chough (*Pyrrhocorax graculus*)	2	173	358
American Crow (*Corvus brachyrhynchos*)	1		369
Carrion Crow (*Corvus corone*)	1	329	
Eurasian Magpie (*Pica pica*)	2	122	184
Hooded Crow (*Corvus cornix*)	2	292, 423	
Little Raven (*Corvus mellori*)	1	362	
Rook (*Corvus frugilegus*)	1	292	
Western Jackdaw (*Coloeus monedula*)	1	292	

**Table 11 animals-11-03226-t011:** List of articles reporting different kinds of human-crow conflicts (*n* = 15). The reference numbers in the table refer to the individual publication numbers given in Appendix A.

Species	*N*	Fecal Droppings	Fouling of Infrastructure	Noise	Aggressive Behavior	Scavenging on Garbage
American Crow (*Corvus brachyrhynchos*)	1	133	133	133		
Blue Jay (*Cyanocitta cristata*)	1		28	28	28	
Carrion Crow (*Corvus corone*)	1		388	388	388	
Hooded Crow (*Corvus cornix*)	2			332, 210	332, 210	210
House Crow (*Corvus splendens*)	2	15, 54	15, 54	15, 54	54	15
Jungle Crow (*Corvus macrorhynchos*)	6	152	152, 388	152, 388, 217	388, 217	152, 217, 404, 407, 411
Rook (*Corvus* *frugelugus*)	2	213		120, 213		
Torresian Crow (*Corvus orru*)	1			119		

**Table 12 animals-11-03226-t012:** List of articles reporting on ecosystem services provided by urban corvids (*n* = 16). The reference numbers in the table refer to the individual publication numbers given in Appendix A.

Species	*N*	Ecosystem Services	
		Seed Dispersal	Health-Based Services (Role as Biosensors)
American Crow (*Corvus brachyrhynchos*)	3		183, 235, 386
Australian Raven (*Corvus coronoides*)	1		313
Blue Jay (*Cyanocitta cristata*)	2	88, 174	
Carrion Crow (*Corvus corone*)	3	391	164, 322
Eurasian Jay (*Garrulus glandarius*)	1	155	
Eurasian Magpie (*Pica pica*)	5	139	164, 322, 100, 424
Hooded Crow (*Corvus cornix*)	1	139	
Jungle Crow (*Corvus macrorhynchos*)	1	391	
Little Raven (*Corvus mellori*)	1		34
Rook (*Corvus frugelugus*)	2	85, 198	
Western Jackdaw (*Coloeus monedula*)	1		424

## Data Availability

Reference list of all publications used in this review is available in Appendix A.

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
