# Peer review of "Corvids in Urban Environments: A Systematic Global Literature Review"

_animals, 2021, doi:10.3390/ani11113226_

Round 1

Reviewer 1 Report

This is study is a comprehensive review on the Corvids and serves as an important work synthesizing some of the factors governing urban corvids and their exploitation of human-dominated landscapes. I enjoyed reading the paper and commend the authors for their efforts in extracting pertinent information from such diverse articles. The paper identifies gaps in our understanding of urban corvids and the road forward. My experience is with House Crows in Asia and the authors pinpoint the need for studies of this widespread species in their native Asia where they have proliferated in cities. I recommend this article for publication, with the following minor comments:

  1. please state the criteria for screening and determining eligible articles in the methods.
  2. please check the numbering of the references as the numbers often do not match. e.g. Ref 54 is listed under House crows but it is about ravens in California.

Author Response

This is study is a comprehensive review on the Corvids and serves as an important work synthesizing some of the factors governing urban corvids and their exploitation of human-dominated landscapes.

>> Thank you!

  1. Please state the criteria for screening and determining eligible articles in the methods.

>> We appreciate the suggestion, however, since we followed the PRISMA guideline (supplementary material Table S1), we believe we have determined the criteria in the Prisma workflow table we submitted as a supplementary material accordingly. In addition to the information given in the Table S1, we have stated criteria for including papers in our data in lines 132-155.

  1. Please check the numbering of the references, as the numbers often do not match. e.g. Ref 54 is listed under House crows but it is about ravens in California.

>> Thank you for pointing this out. The reviewer is correct, and we have made the necessary corrections.

Reviewer 2 Report

This paper is interesting to read and answers a novel question that is pertinent to how animals are affected by anthropogenic influences. I believe that there is merit in publishing this work but not in its current form. The article needs to be written more succinctly and have clearer tables and figures that allow the reader to pull out the important findings from them more easily.

Please re-write the simple summary, as this is far too long and complicated (using many technical terms) to be understandable to the non-scientist. 

Please consider some turns of phrase in the manuscript, e.g. "global winners" is quite anthropomorphic. 

Please check the written English to ensure that sentences flow and make grammatical sense. For example, Line 76 should be "The corvids are..."

Please provide a more fluent bridge between the introductory paragraphs on urbanisation and what it is, and the reasons for the focus on corvids. Currently, there is quite the leap between the first two paragraphs the explanation of "why corvids were studied". 

Please define key terms or technical phrases for the reader, e.g. syanthropic.

The methods are detailed and clear. It's great to see the PRISMA guidelines for meta-analyses being followed. Clear explanation of key terms and how these were decided upon has been provided to enable the methods to be repeatable. 

The result section is far too long and contains many tables that add little value in terms of the results they show. Please re-write this section, removing superfluous or descriptive information and presenting summaries of findings (e.g. means, medians, modes, percentages or proportions of the subjects of the papers that have been identified for review). 

Could table 1 be included in supplementary information? Does it provide anything useful for inclusion in the main text? Perhaps simply state the species most commonly studied as a proportion of the dataset?

Figure 2 is hard to read. Please can you re-draw this as a more professional style graph?

Lots of the tables do not really contain results per se, but are simply signposts to where the information occurs in the literature. Please pull out some data from these papers, for example the number or percentage of papers that show an increase in population sizes, for example.

Terms such as ecosystem services need to be explained in the context of what was investigated in the review. All species have a role in ecosystem services so what do you mean here?

The results section could be improved overall by replacing many of the tables with a couple of clear figures that indicate trends in corvid publications visually. This could then be supported by some inferential testing, perhaps a logistic regression (?) that could be used to predict the likely direction of future research based on any trends identified. 

The discussion is also long and the introductory paragraphs to it need to be rationalised. Explain the key findings from the research and then evaluate their wider importance. Many areas of the discussion are devoid of outside referencing and therefore it is hard to judge the quality of the information presented and how it links to that published on. For example, on Line 627 you state that the Siberian jay (and influences of urbanisation) would be interesting to study but you give no reasons for why. Perhaps this is down to the ecology of the species and its behavioural plasticity? 

Is the section on limitations needed? Could you take this section on limitations and rephrase as future areas for research extension? Where would you take this research in the future and why? 

Please re-work the discussion to provide more focus on trends and patterns and your dataset (inclusion of some inferential testing would help with this) and this would then, overall, help you better frame your conclusions. 

Author Response

General comments:

This paper is interesting to read and answers a novel question that is pertinent to how animals are affected by anthropogenic influences. I believe that there is merit in publishing this work but not in its current form. The article needs to be written more succinctly and have clearer tables and figures that allow the reader to pull out the important findings from them more easily.

>> Thank you! We greatly appreciate the comments given the reviewer, and we have tried to follow your suggestions as detailed as possible. We agree that our article is a little bit long. However, we want to note two topics; 1) as the Animals is an online Journal, there is no length restrictions; 2) our MS is a review type article that are normally longer than normal, basic research articles. Related to the tables, we want to keep all of them in the same format, so that the readers can e.g. easily find the original articles that were reporting specific results given in the tables.

Specific comments

  1. Please re-write the simple summary, as this is far too long and complicated (using many technical terms) to be understandable to the non-scientist.

>> Thank you for your important suggestion. The reviewer is correct. We have made the necessary changes accordingly.

New, edited Simple Summary:

“With regard to their high adaptability to human settlements and global distribution, corvid birds (crows, ravens, jays etc.) are good models to understand the impacts of urbanization on wildlife. Here we reviewed scientific publications on the impacts of urbanization on corvids. At least 30 corvid species were found to have successfully adapted to urban environments. The majority (72%, of a total of 424 articles) of the studies reported positive effects of urbanization on corvids. The availability of easily accessible food and nesting sites, coupled with low levels of predation, were found to be the most important factors benefitting corvids in cities around the world. Studied topics ranged from population size and density, breeding biology, and nesting site selection to control and management of Corvidae in cities. Despite biases in the distribution of the reviewed papers, our review attests that corvids have demonstrated high levels of adaptability to urban environments across space and time.”

  1. Please consider some turns of phrase in the manuscript, e.g. "global winners" is quite anthropomorphic.

>> Thank you for such input. We agree with the reviewer’s point and we have made the necessary changes by using more appropriate terminology, therefore we have chosen to use “Urban exploiter” term base on the definition of birds thriving in cities by Kark et al., 2007.

  1. Please check the written English to ensure that sentences flow and make grammatical sense. For example, Line 76 should be "The corvids are..."

>> Thank you for highlighting such mistake. Also, other reviewers suggested to do so. The Reviewer is correct. We have made the necessary corrections.

  1. Please provide a more fluent bridge between the introductory paragraphs on urbanization and what it is, and the reasons for the focus on corvids. Currently, there is quite the leap between the first two paragraphs the explanation of "why corvids were studied".

>> Thank you for the suggestion. We have now revised the two first paragraphs as follows:

“Urbanization is a spatio-temporal process of the development of cities and the increase of concentration of populations in them followed by a transformation of natural habitats into artificial ones [1,2]. In general, urbanization is strongly associated with increased cover of imperious structures (e.g. buildings, streets) and human population density as well as the fragmentation, degradation and loss of natural habitats. An urban development is an ecological modification that often alters the functions of a given ecosystem by affecting the structure of the food chain by removing or adding species, by encouraging human tolerance and adaptation, by increasing health risks for humans and wildlife, and by modifying ecological processes in relation to ecosystem services [3]. Urbanization leads to complex, diverse systems characterized by high levels of human disturbance, pollution, and landscape and environmental changes [1,2,4]. These changes can affect the biology, behavior, morphology, and reproductive and survival traits of wildlife, and can be responsible for the disappearance of native species and the appearance of non-native ones [5]. Therefore, understanding these effects is essential for successful wildlife conservation and management in urban habitats.

The negative impact of man-made landscapes and infrastructures on wildlife has been detected in many studies [6-8]. However, numerous studies have also described how certain species, like corvids (e.g. crows, magpies), can benefit from these infrastructures, such as using buildings, poles and power lines as nesting sites [9-11] . In addition, antropogenic food resources and milder microclimate in cities might benefit many corvid species [1,2,6]. Urbanization has been considered as an overwhelming evolutionary force acting on the life-history traits and population genetics of species [8]. Currently, urbanization is still expanding at an accelerating pace [12], unfortunately coinciding with a continuous increase in habitat loss. Although, studies on the effects of urbanisation on birds, at a community or individual species level, have been widely conducted, multi-species approaches with species belonging to the same family are still very scarce.”

  1. Please define key terms or technical phrases for the reader, e.g. syanthropic.

>> Thank you for the suggestion. We agree with the reviewer, and we have provided brief definition for such terms in parentheses after each one. 

  1. The methods are detailed and clear. It's great to see the PRISMA guidelines for meta-analyses being followed. Clear explanation of key terms and how these were decided upon has been provided to enable the methods to be repeatable.

>> Thank you!

  1. The result section is far too long and contains many tables that add little value in terms of the results they show. Please re-write this section, removing superfluous or descriptive information and presenting summaries of findings (e.g. means, medians, modes, percentages or proportions of the subjects of the papers that have been identified for review).

>> We agree that our article is a little bit long. However, we want to note two topics; 1) as the Animals is an online Journal, there is no length restrictions; 2) our MS is a review type article that are normally longer than normal, basic research articles. Related to the tables, we want to keep all of them in the same format, so that the readers can e.g. easily find the initial articles that were reporting specific results given in the tables. Therefore, we have not re-written this section. We also want to note that our tables contain the main information that our data search were catch, and therefore, it will be important to give all this information in the tables. Because in some case the table are a little bit long, we think that it might be reader-friendly to give some shorth, basic summary of each table in the main text. We want also to pinpoint, that the other two reviewers were satisfied the structure of the manuscript.

  1. Could table 1 be included in supplementary information? Does it provide anything useful for inclusion in the main text? Perhaps simply state the species most commonly studied as a proportion of the dataset?

>> Thank you for pointing this out. As table 1 summarizes important information on different corvid species studied over time and space and how frequently different species were subjected to such investigation. We believe that such general-level table is necessary for readers to have easy introduction and access to well defined publications for their potential work, and makes it less time consuming for reader to obtain necessary data that are precisely interested in.

  1. Figure 2 is hard to read. Please can you re-draw this as a more professional style graph?

>> Thank you for this suggestion. Also, other reviewers suggested to do so. Indeed, we agree with the suggestion, and we have made the necessary changes.

  1. Lots of the tables do not really contain results per se, but are simply signposts to where the information occurs in the literature. Please pull out some data from these papers, for example the number or percentage of papers that show an increase in population sizes, for example.

>> Thank you for pointing it out. This is the way how data are typically presented in a qualitative systematic review, like in our case. Through this review, our goal is to provide readers a single document summarizing, as much as possible, the knowledge we current have about corvids in urban settlement. Accordingly, table 2 was produced in order to highlight the effect of urbanization on corvids population’s sizes while demonstrating the proportion of papers discussing such topic. As, We opted for a systematic review in which different results of different studies were summarized, by making such tables we aimed to facilitate readers access to published information by specifically organizing the reviewed publications according to the topic in question as well as a species studied in the different addressed topics so far.

  1. Terms such as ecosystem services need to be explained in the context of what was investigated in the review. All species have a role in ecosystem services so what do you mean here?

>> Thank you for pointing it out. Within our review, we aim to shed lights on the potential positive role of Corvids in urban areas, rarely explored but still investigated at some point. Thusly, within the manuscript, as suggested by the reviewer, we have described what type of “ecosystem services” observed in different corvid species so far as follows:

“although corvids are often perceived as a nuisance, recent studies shed light on their potential role to provide supporting ecosystem services, like helping seed dispersal of fegaceous tree species through transporting fruits and seeds per caching trips over long distances [158,159]., and regulation ecosystem services like health-based services by acting as a surveillance tool of important zoonotic pathogens and by reducing ani-mal remains by scavenging within urban settings [23,36, 37,160,161]”.

  1. “The results section could be improved overall by replacing many of the tables with a couple of clear figures that indicate trends in corvid publications visually. This could then be supported by some inferential testing, perhaps a logistic regression (?) that could be used to predict the likely direction of future research based on any trends identified.”

>> Thank you for pointing it out. In this review, we want to be consistent to show our data and results between all the topics that we have considered. Other two reviewers liked the structure a lot. If we will replace tables by figures, then we will lose the possibility to cite original papers reporting the observed results, and that will be not reader-friendly way to report our data and results.

-Related to testing, our review is a systematic review, not e.g. a Meta-analysis type of review, in which such kinds of statistical testing are necessary. Therefore, we have not performed any statistical testing (indeed, our basic data does not make it possible). However, this type of meta-analysis might be considered as a future research topic. Therefore, we suggested this kind of work for the future work.

  1. Comment from Reviewer 2 suggestion that “The discussion is also long and the introductory paragraphs to it need to be rationalised.”

>> Thank you for the suggestion, we agree with the recommendation and we have revised and rationalized the text, and shortened the beginning of the discussion section accordingly.

“Our results indicate that corvids have been widely successful in adapting to urbanized environments over recent decades over a worldwide geographic scale. At least 23% of 130 corvid species in the world were reported to live in urbanized environments. With a majority of the reviewed articles reporting positive effects of urbanization on the population density or abundance of corvid species and a majority reporting higher breeding success, attesting thus this group’s extreme flexibility in resource use, and highly opportunistic and plastic behavior, often enabling them to reach high abundances in various cities around the world. In contrast, we found no evidence of occurrence in urban environments for 100 corvid species. However, this does not mean that these species avoid cities and towns; rather it reflects the lack of detailed studies documenting the occurrence and breeding of corvids in urban environments. Accordingly, while the establishment and colonization of corvids in urban environments have drawn interest from researchers since the early 1990s, there has been a recent surge of scientific interest in corvids in urban environments in many areas of the world, and research on urban corvids is a rapidly emerging field of inquiry.”

  1. “Many areas of the discussion are devoid of outside referencing and therefore it is hard to judge the quality of the information presented and how it links to that published on”.

>> Thank you for pointing it out. We have used both the papers cited in the tables as well as some additional references when appropriate in the discussion. However, since we reviewed specific articles related only to corvids in urban areas, we found it necessary to discuss our findings in a broader way, which called for the employment for further information outside of the targeted topic box. However, we have added some extra citations so that it is now easier to judge the quality of the information presented, and how our discussion is related to the published literature.

  1. “On Line 627 you state that the Siberian jay (and influences of urbanization) would be interesting to study but you give no reasons for why. Perhaps this is down to the ecology of the species and its behavioural plasticity?” 

>> We highly appreciate the reviewer’s suggestion. We have made the necessary additions as suggested.

“Evidently, habitat needs of some corvid species are not fulfilled in cities. For example, the Siberian Jay is an old-growth bird species living in old-aged and large-sized coniferous forests [182], therefore this species does not breed or over-winter in cities, since urban green areas are normally quite small-sized, fragmented and deciduous tree dominated [183].”

  1. Is the section on limitations needed? Could you take this section on limitations and rephrase as future areas for research extension? Where would you take this research in the future and why?

>> While we highly appreciate the reviewer’s suggestion. However, in this kind of review type studies, it is very important to highlight the different limitations of our review so that the readers would be aware of the exact information is reviewed, especially when identifying potential biases expressed along the way. Indeed, most if not all studies have limitations, and in our mind, it is important to always report them openly. While we definitely agree with the reviewer’s suggestion about discussing future areas of researches and research extensions, we do believe that we have addressed such ideas to some extent in the “Research needs and avenues” section, following the Limitation section.

  1. Please re-work the discussion to provide more focus on trends and patterns and your dataset (inclusion of some inferential testing would help with this) and this would then, overall, help you better frame your conclusions.

>> We highly appreciate the reviewers’ recommendations, as mentioned in our comments above. Statistical testing is outside of the scopes of the systematic reviews. However, we have tried to frame our conclusions a little bit.

Conclusion:

Our review shows that corvids have long been associated with the development of urban environments, and their worldwide distribution make them the perfect model system to study the effects of urbanization on wildlife. A considerable proportion of species in the Corvidae family have already been shown to adapt to urban environments and, with consideration to the geographical and taxonomical bias in our literature sample, it appears likely that many more species will be shown to be successful in the adaptation process. The primary traits of corvids that enables them to exploit new, urban environments are their high behavioral plasticity and flexible resource use. With easily accessible food being the most important resource attracting these birds into cities; influencing thus different traits of habitat selection (e.g. use of new nesting sites) and life history (e.g. earlier nesting, larger clutches, higher fledging success, reduced home ranges and territoriality) as well as behavior (increased tolerance of humans); understanding the relative importance of these changes in each species will be fundamental to better understand the adaptation process and human-wildlife interactions and to develop efficient management applications. While the effects of urbanization on numerous corvid species have been relatively well explored, there are important gaps in our knowledge, calling for exercise of a more diversified approach to study this process with different, complementary methods, and a focus on the potential benefits of this process such as the ecosystem services that corvid species provide. We encourage researchers to also address these aspects for a more balanced view of corvids in urban environments.”

Reviewer 3 Report

General comments:

This is a great review of corvids in urban environments. I just have a few suggestions that should be considered before this can be accepted for publication.

Specific comments:

Line 76: Corvids is not the family. Please change to Corvidae.

Line 144: I think there is a comma missing…

Lines 154-155: Why not just include this information in topic 3 above? That way, you don’t have to add this clarifying statement here.

Figure 2: Why is this information displayed as a figure and not a table like all the other results? If this stays as a figure, I think it needs to be edited. First, the 3-D nature of the figure is not needed. Second, delete the gray background boxes. Third, make the bars bigger (you can move the figure legend to make room for this. Fourth, the x-axis and y-axis labels do not need to be in all caps. I would try to match the style of Figure 1 (except the gray lines and use a sans serif font for both).

Lines 258-260 and Table 8: Are the pathogens to which you are referring really of human origin? I would say zoonotic and delete mention of human origin since West Nile virus and tick-borne pathogens, for example, are not of human origin. You could also say “of human importance” on line 259.

For all the tables, I would suggest putting the number of studies along with references in parentheses, instead of just the references. It would help so that readers would not have to count all the references in a given table.

Line 493: I would not refer to birds as vectors. Typically, the term vector is used to describe arthropod transmitters of disease. This can be deleted. I would say reservoirs and spreaders of pathogens.

Line 522: Health is spelled wrong.

Author Response

General comments: This is a great review of corvids in urban environments.

>> Thank you!

Specific comments

  1. Line 76: suggest change “Corvids” to “Corvidae”.

>> Changed as suggested.

  1. Line 144: suggest “a comma missing”…

>> Added as suggested.

  1. Lines 154-155: Why not just include this information in topic 3 above? That way, you don’t have to add this clarifying statement here.

>>Thank you for your suggestion. We moved these sentences (We also included studies about specific biological or physiological parameters as these can be important to explain the birds’ responses to human-155 caused changes in their environment) under the Point 3 above.

  1. Figure 2: Why is this information displayed as a figure and not a table like all the other results?

>> Thank you for this suggestion. The reason behind choosing to display the information in a figure instead of a table is that we believe that it would be easier and more comprehensive for readers to compare in this specific topic (urbanization effect) the frequency in which positive, negative and no effect of urbanization on a given corvid species were discussed in literature. Displaying such information in a figure can be more attractive and easily on the eye than when displayed on a table.

  1. If this stays as a figure, I think it needs to be edited. First, the 3-D nature of the figure is not needed. Second, delete the gray background boxes. Third, make the bars bigger (you can move the figure legend to make room for this. Fourth, the x-axis and y-axis labels do not need to be in all caps. I would try to match the style of Figure 1 (except the gray lines and use a sans serif font for both).

>> Thank you for pointing that out. The reviewer is correct, and we have made necessary the changes.

  1. Lines 258-260 and Table 8: Are the pathogens to which you are referring really of human origin? I would say zoonotic and delete mention of human origin since West Nile virus and tick-borne pathogens, for example, are not of human origin. You could also say “of human importance” on line 259.

>> Thank you for pointing that out. The reviewer is correct, and we have made necessary the changes.

  1. For all the tables, I would suggest putting the number of studies along with references in parentheses, instead of just the references. It would help so that readers would not have to count all the references in a given table”

>> Thank you for the suggestion. We agree with the reviewer and we added an extra column in each table containing the count of all references per species.

  1. Line 493: Delete the word vectors (use the word “spreader” or “reservoirs”)

>> Changes made as suggested.

  1. Line 522: Typo – change helt to “health”

>> Corrected as suggested.

Round 2

Reviewer 2 Report

The authors have considered all reviewer points and have amended their article accordingly. I feel that this is suitable for publication with a few small edits for clarity. 

I think that moving Table 1 to supplementary information would really help reduce the length of the article. This table simply shows the papers that were collected but you have provided more relevant information in the graphs and further tables. Perhaps have a short paragraph that explains the most commonly studied species and most commonly published on topic. 

Please check your spelling and grammar, for example the Y axis on Figure 1 should be Number 

Similarly for Figure 2, Suggest number of articles for the Y axis. 

I would explain further how positive, negative and neutral has been decided upon so the reader can see what these categories mean.

Scientific names, wherever they appear in the text, need to be italics.

Ignoring information in tables and in figures, scientific names are only included the first time a species is included in the text. There are scientific names appearing multiple times when this is not needed as they have already been quoted. Please go back and check through to see when the species is first mentioned (that is when the scientific name should be quoted). 
